# Mediterranean Diet Effect on the Intestinal Microbiota, Symptoms, and Markers in Patients with Functional Gastrointestinal Disorders

**DOI:** 10.3390/microorganisms12101969

**Published:** 2024-09-28

**Authors:** Elena Garicano Vilar, Sara López Oliva, Bruno F. Penadés, Guerthy Melissa Sánchez Niño, Ana Terrén Lora, Sara Sanz Rojo, Ismael San Mauro Martín

**Affiliations:** Research Centers in Nutrition and Health (CINUSA Group), Paseo de la Habana 43, 28036 Madrid, Spain; elena@grupocinusa.es (E.G.V.); sara@grupocinusa.es (S.L.O.); estadistica@grupocinusa.es (B.F.P.); melissa@grupocinusa.es (G.M.S.N.); ana.terren@grupocinusa.es (A.T.L.); sara.sanz@grupocinusa.es (S.S.R.)

**Keywords:** Mediterranean diet, microbiota, dysbiosis, fungi, bacteria, zonulin, gastrointestinal disorders

## Abstract

The Mediterranean diet (MD) has beneficial effects on the intestinal microbiota by the promotion of bacteria associated with a healthy gut. However, its impact on intestinal fungi, among others, is still unknown, and how it affects digestive symptoms and different biomarkers in patients with gastrointestinal (GI) disorders has hardly been explored. The present study evaluated the effect of the MD on gut microbial diversity and structure and intestinal symptoms and biomarkers after 6 weeks of dietary intervention in 46 patients with GI disorders. Dysbiosis in fungal composition and diversity was observed, with a significantly lower abundance of Sordariomycetes, Leotiomycetes, and Orbiliomycetes; a significantly higher abundance of Saccharomycetes; the Chytridiomycota and Mucoromycota phyla were significantly reduced; and the bacterial microbiota remained unchanged. In addition, various GI disorders decreased and associations between stool consistency and intestinal permeability were found with the bacterial genera *Alistipes* and *Roseburia*. Thus, the data suggest that MD can alter the fungal intestinal microbiota and improve GI disorders.

## 1. Introduction

The human microbiome is a community of microorganisms found on or within human tissues and biofluids, especially in the mouth, intestines, vagina, and skin; however, the highest concentration of microorganisms is found in the gastrointestinal tract [1]. Therefore, it is important to understand how specific communities of microorganisms and their diversity influence the human body and disease [2].

One factor influencing microbiota composition is diet, which has been an attractive target for clinical research [3]. The Mediterranean diet (MD) contains 3–9 servings of vegetables, 1–2 servings of fruits, approximately 13 servings of cereals, and up to 8 servings of olive oil per day. In terms of its quantified nutrient intake, the MD per day contains approximately 2222 kcal, 33 g of fiber, and 37% total fat, of which 18% is monounsaturated and 9% is saturated [4]. It is well known that the MD has beneficial health effects such as reducing the risk of obesity, cardiovascular disease, liver diseases (e.g., fatty liver), diabetes mellitus, metabolic syndrome, several types of cancer, and neurodegenerative disorders [5,6,7,8,9,10,11,12].

Recently, several studies have shown that the MD also affects microbiome diversity and modulates it [13,14]. A higher level of adherence to the MD has been associated with changes in the gut microbiome composition, and this is characterized by a reduction in the abundance of Proteobacteria and an increase in the production of short-chain fatty acids (SCFAs) [15]. The precise mechanism through which greater adherence to the MD has favorable effects is unknown. However, there is evidence that demonstrates five adaptations induced by the MD: reduction in lipid levels; protection against oxidative stress, inflammation, and clotting; alteration of hormones and growth factors linked to cancer development; the blocking of nutrient detection pathways by restricting specific amino acids; and the production of metabolites such as acetate, propionate, or butyrate (which impact metabolic well-being through the intestinal microbiota) [16].

The impact of diet on the gut microbiome has been studied using metagenomic data, which demonstrate that specific nutrients, especially proteins and insoluble fibers, affect gut function, structure, and the secretion of metabolites that modulate immune function [17]. Other beneficial effects of microbiota metabolites, such as SCFAs, ω-3 fatty acids, succinate, or kynurenic acid are mediated by specific G protein-coupled receptors that are expressed on enteroendocrine and immune cells [18]. Therefore, long-term adherence to certain dietary patterns may have a greater impact on the taxa composition and diversity of the gut microbial population than short-term dietary modifications. In addition, long-term consumption of plant-rich diets with restricted caloric intake has been associated with more phylogenetically varied microbiota [19].

The study of microbial interactions focuses mainly on bacteria, whereas fungi are generally considered minor components. For example, recent rapid sequencing efforts have suggested that fungi make up approximately 0.1% of microorganisms in the body [20,21]. However, this quantity could be higher since there are more than 184 species of mycobiota colonizing the human gut, with *Candida*, *Saccharomyces*, and *Cladosporium* being the most abundant [22]. Some studies have suggested that fungal populations are more variable and may be influenced by fungi in the environment and that bacteria are more abundant than fungi; consequently, bacterial communities are more stable than fungal communities [23]. The intestinal fungi composition can be modulated by age, psychological states, or usual diet. Fungal communities change between early to late pregnancy, with *Mucor* during early pregnancy being a risk factor for gestational diabetes mellitus and macrosomia [24]. Other studies have shown that gut mycobiota help to maintain intestinal homeostasis and systemic immunity, and they also have an active role in inflammatory bowel disease (IBD), irritable bowel syndrome, colorectal cancer, liver diseases, and metabolic and neurological disorders [25,26,27,28]. To modulate fungal microbiota composition, diet changes can be also an effective tool to affect fungal microbiota. The consumption of plant-based diets has been linked to an increase in *Candida*, while the consumption of animal-based diets has facilitated the expansion of *Penicillium* species [13]. While previous studies have listed changes in the microbiota in relation to diet, few studies have demonstrated how the MD can affect the composition and diversity of the fungal microbiota and its interaction with bacteria.

In addition, intestinal permeability can also affect gut microbiota composition. Zonulin is a protein that regulates intestinal membrane permeability [29]. In fact, several health problems could derive from high levels of zonulin, such as some gastrointestinal diseases (food intolerances, irritable bowel syndrome, or inflammatory bowel disease) that have a variety of gastrointestinal symptoms associated with them [30]. The main objective of this study was to evaluate the effects of the MD on the composition and diversity of bacterial and fungal gut microbiota, digestive symptoms, blood biomarkers, and intestinal permeability after six weeks of dietary intervention in Spanish patients with gastrointestinal symptomatology.

## 2. Materials and Methods

### 2.1. Study Cohort

The study cohort consisted of 46 Spanish participants between the ages of 25 and 64 (41.39 years ± 8.94) who presented at least two digestive system symptoms. The characteristics of participants in the study cohort are listed in Table 1. The inclusion criteria were an age between 18 and 65 and a diagnosis of gastrointestinal symptoms (bloating, abdominal pain, flatulence, diarrhea, borborygmi, decreased stool consistency, constipation, nausea, belching, acid regurgitation, and epigastric pain). The exclusion criteria were multiple chemical sensitivity, diet follow-up, and chronic diseases, such as chronic obstructive pulmonary disease, cancer, lupus, AIDS, heart attack, hepatitis, tuberculosis, sclerosis, colectomies, enterectomies, ulcerative colitis, Crohn’s disease, celiac disease, and wheat allergy.

### 2.2. Study Design

An ecological study was conducted to determine the effects of the MD on bacterial and fungal microbiota in the intestine of a human study cohort with intestinal symptoms (Figure 1). This study was conducted by the Research Centers in Nutrition and Health (CINUSA Group, Madrid, Spain) at Clínica CINUSA and Ruber International Hospital (Madrid, Spain).

### 2.3. Ethical Considerations

The study was approved by the Ethics Committee for Research with Medicines (*Comité Ético de Investigación con Medicamentos*) of the Madrid Ruber International Hospital (QuirónSalud Group, Madrid, Spain) (Sa-16,151/19—EC: 393; CIN-GLU-01-19).

The study was conducted in compliance with the latest version of the Declaration of Helsinki (World Medical Association, Fortaleza, Brazil 2013), good clinical practice standards (ICH 2016 R2), and legal standards and regulations for biomedical research in humans (Law 14/2007 and royal decree 1090/2015).

In addition, the study was conducted under strict compliance with the European General Data Protection Regulation 2016/679 of the European Parliament and EU Regulation 2016/679 of 27 April 2016, on data protection of natural persons (RGPD) and Organic Law 3/2018 (LOPD).

All participants took part in the study voluntarily. Only those who met the selection criteria and signed the informed consent form were part of the study.

### 2.4. Biomarker Analysis

Blood samples were obtained from all study participants for biochemical analyses. Iron, ferritin, transferrin, and CRP (C-Reactive Protein) were selected as biomarkers in this study. Additionally, fecal zonulin was used as a biomarker of intestinal permeability and was quantified using human zonulin ELISA kit (Novus BiologicalsTM NBP3-21146) at the beginning and end of the intervention by oral administration of non-metabolizable substances [31,32].

### 2.5. Dietary Guidelines

The dietary guidelines were designed by standardizing the diet and eating habits of the participants. Semi-personalized menus were designed according to food guidelines and the definition of the MD [5,33,34]. They were adapted to the number of meals per day, different tastes and preferences, and were rotated every three weeks. General recommendations, such as not consuming alcohol, drinking water as the main source of hydration, and not drinking soft drinks, snacks, processed baked goods, or ultra-processed foods, were also included.

The Mediterranean Diet Adherence Screener (MEDAS) [35] questionnaire was used to assess adherence to the MD before the intervention. The MEDAS questionnaire consisted of 14 dichotomous questions (yes = 1, no = 0) depending on whether participants adhered to each component of the MD (1 point) or not (0 points). The final MEDAS scores ranged from 0 to 14. For categorization of the adherence to the MD we applied the following criteria: weak/moderate adherence, ≤9; good or very good adherence >9.

### 2.6. Molecular Analysis

#### 2.6.1. DNA Extraction

DNA from fecal samples was extracted using the PSP Spin Stool DNA Plus Kit (Invitrogen Molecular GmbH, Belin, Germany), following the manufacturer’s instructions.

#### 2.6.2. DNA Quality

Quant-iT PicoGreen ™ dsDNA Reagent (Invitrogen ™, Thermo Fisher Scientific, Massachusetts, USA), an ultrasensitive fluorescent nucleic acid stain, was used to quantify the double-stranded DNA (dsDNA). Fluorescence emission intensity was measured at 260 nm using a VICTOR3 spectrofluorometer. The Quant-iT PicoGreen ™ (Invitrogen ™, Thermo Fisher) assay remained constant in the presence of several compounds that commonly contaminate nucleic acid preparations.

#### 2.6.3. DNA Library Construction

The sequencing library was prepared by random fragmentation of the DNA sample, followed by the ligation of primer 5′ and 3′. The fragmentation and binding reactions were combined in a single step, which greatly increased the efficiency of the library preparation process. Fragments bound to the adapter were amplified using PCR and purified on a gel.

#### 2.6.4. Sequencing

For cluster generation, the library was loaded into a flow cell, where fragments were captured on a glass of oligos attached to the surface and complementary to the library adapters. Each fragment was amplified into different cloned clusters using bridge amplification. After cluster generation, the templates were sequenced using high-throughput Illumina MiSeq sequencing to examine the bacterial and fungal gut communities following PCR amplification of the V4 region of 16S rRNA gene and ITS.

### 2.7. Bioinformatic Analysis

The USEARCH v11.1 [36] sequence analysis tool was used for bioinformatics analysis of the sequenced samples. The software analyzed the sequences of the 16S (bacteria) and ITS (fungi) extensions using several heuristic computational algorithms to determine the ZOTUs present in the sample and estimate their abundance in each sample and their corresponding taxonomic annotation. The quality of the sequences was evaluated prior to bioinformatics analysis.

### 2.8. Statistical Analysis

The assumption of normality of the independent samples was verified using the Shapiro–Wilk test, as well as the differences between the paired samples. The assumption of symmetry was verified for paired samples using a typified version of the asymmetry index. The means were compared using a Student’s *t*-test and the sign test was opted for only when the interpretation of the results for paired samples was at risk in the presence of strong asymmetries in the distribution of differences.

Alpha diversity was analyzed using the Chao1 and Shannon indices. Beta diversity was analyzed via the permutational multivariate analysis of variance (PERMANOVA), representing the results with principal coordinate analysis (PCoA), and linear regressions were performed. Differential abundance analysis was performed using multivariate associations of the microbiome with linear models (MaAsLin2) [37]. The multivariate associations between fungal composition and the effect of MD were examined, proposing a mixed-effects model in which time, age, BMI, and sex were included as fixed effects and subjects as random effects. Relative abundance was calculated, excluding non-informative taxa, and values were normalized using total sum scaling (TSS) and transformed using arcsine square root transformation (AST), with a minimum prevalence of 10%. The significance level was corrected for multiple comparisons (FDR, False Discovery Rate), and the adjusted significance level was determined at 0.05 for the time variable, and 0.15 for the covariates. Finally, a heatmap and correlogram were used to search for associations among gastrointestinal symptoms, biomarkers, and sample characteristics using Spearman’s test.

Taxa without taxonomic assignment were excluded from the analysis in all tests, except for the alpha and beta biodiversity analyses (whose analysis were performed with the microeco package), and the contrasts were bilateral, with a 95% confidence level. All analyses were performed using R software v4.3.2 (RStudio, Inc., Boston, MA, USA) for statistical computing and graphics. The R package microeco v0.12.0 [38] was used to analyze the microbial communities.

## 3. Results

The number of starting ZOTUs in the ITS amplicon was 1340 (5,186,964 reads). After filtering the ZOTUs by very low abundance (n ≤ 10; 94 ZOTUs), 1246 ZOTUs and 5,186,524 effective reads were obtained. As for the 16S amplicon, 1647 ZOTUs (3,629,451 reads) were found after excluding sequences associated with non-prokaryotes (1 ZOTU), and 1608 ZOTUs and 3,629,222 effective reads were obtained after excluding very low abundance ZOTUs (n ≤ 10; 29 ZOTUs). The amplicon sequence data were not rarefied because the sequencing depth was consistent across all samples.

The taxonomic classification was very poor at levels below the class in the ITS amplicon. Although phylum and class levels reached an average of 60%, the remaining levels were significantly lower, being less than 20% in most cases, and less than 10% for genera. This fact was decisive in undertaking the analysis, and the decision was made to reach the fungal classes at the highest level, while there were not enough reads and representativeness evidenced in the remaining levels. In contrast, there was low data loss for the 16S amplicon, where the taxonomic genera assignment was close to 80%.

### 3.1. Fungal Diversity and Composition

The first step was to explore the alpha diversity indices before and after intervention. Richness was reduced post-MD according to the Chao1 index (paired *t*-test, *p* < 0.0001) (Figure 2A, left). Also, richness and evenness were reduced post-MD according to the Shannon index (paired *t*-test, *p* < 0.0001) (Figure 2A, right). In addition, PERMANOVA detected that the spatial distribution of fungal communities between time points was significantly separated (*p* = 0.001) (Figure 2B).

Regarding the relative abundance of fungal phyla and classes, the data revealed that the phylum Ascomycota dominated the intestines of pre- and post-MD participants, unlike Chytridiomycota and Mucoromycota, which disappeared after diet intervention (Figure 2C, left). Among fungal classes, Saccharomycetes predominated in pre-MD patients, followed by Sordariomycetes and Eurotiomycetes (Figure 2C, right). However, Sordariomycetes were negligible and the presence of Saccharomycetes increased post-MD. 

These same data were used to analyze whether the differential abundance of fungi changed significantly in the gut of post-MD participants using the two taxonomic levels with the highest taxonomic allocation (phylum and class) and a mixed-effects model with time, age, BMI, and sex as fixed effects and subjects as random effects. The results showed that the abundance of Chytridiomycota and Mucoromycota phyla decreased significantly after dietary intervention (FDR < 0.05) (Figure 2D,E). Sordariomycetes, Leotiomycetes, Rhizophydiomycetes, Spizellomycetes, Umbelopsidomycetes, and Orbiliomycetes classes also decreased or became extinct, except for Saccharomycetes whose abundance increased (FDR < 0.05) (Figure 2F–L). We also considered whether the covariates introduced in the model offered significant results, making the FDR more statistically liberal. Fungal abundance differed according to sex; Eurotiomycetes and Leotiomycetes were more abundant in men and Saccharomycetes in women (FDR < 0.15) (Figure 2M–O).

Based on the same variables introduced in the previous model, the scores of the pre-MD MEDAS questionnaire were analyzed to check for differential abundance based on adherence to the MD before the patients started dietary intervention. To maximize detection of potential effects, MEDAS scores were also analyzed after being categorized as ≤9 low and >9 high. No taxa were associated with adherence to the MD even after increasing the FDR significance limit to 0.25.

On the other hand, the correlation slope differed (*p* < 0.0001) between Ascomycota and Basidiomycota at the different time points (Figure 2P). There was a significant post-MD decrease in the Basidiomycota/Ascomycota ratio (sign test, *p* < 0.05) (Figure 2Q), despite the striking number of outliers that expanded the presence of Basidiomycota in the intestine. And we also examined whether Chao1 richness varied according to the adherence to the MD reported by MEDAS. On one hand, a positive trend was observed (R^2^ = 0.07; *p* = 0.074) (Figure 2R, left); and the comparison was significant once the MEDAS scores were categorized (unpaired *t*-test, *p* = 0.024) (Figure 2R, right).

### 3.2. Bacterial Diversity and Composition

The biodiversity of species assessed using the proposed alpha diversity indices revealed no post-MD changes in the Chao1 (paired *t*-test, *p* = 0.379) (Figure 3A, left) or Shannon indices (paired *t*-test, *p* = 0.559) (Figure 3A, right). In addition, the distance matrix of bacterial communities almost entirely overlapped between the times (*p* = 1) (Figure 3B).

Phyla relative abundance data showed that Firmicutes and Bacteroidetes dominated the bacterial intestinal microbiota of pre-MD participants and that these phyla continued to lead after the MD, with a slight increase in the abundance of Bacteroidetes at the expense of Firmicutes (Figure 3C, left). Among the bacterial genera, *Bacteroides* was predominant, followed by *Faecalibacterium, Alistipes, Prevotella*, and *Eubacterium*, which continued with slight variations after six weeks of intervention (Figure 3C, right).

A model was generated to analyze bacterial abundance following the same steps as the ITS amplicon, but including the remaining taxonomic levels to have a satisfactory taxonomic assignment for all. The results did not show any significant association with time or any of the covariates for the taxa present at different taxonomic levels (FDR > 0.05). Similarly, no differences were observed in the numerical and categorical values of MEDAS scores when added to the model with bacterial taxa.

Furthermore, the relationship between Firmicutes and Bacteroidetes between the time points was similar (*p* = 0.626) (Appendix A), and there was a non-statistically significant (sign test, *p* = 0.096) decreasing trend of post-MD values in the Firmicutes/Bacteroidetes ratio (Appendix A). And regarding adherence to MD, it was found that it did not vary sufficiently with the Chao1 richness (R^2^ = 0.004; *p* = 0.671) (Appendix A) and that the categorized scores also did not change in this regard (unpaired *t*-test, *p* = 0.946) (Appendix A).

### 3.3. Relationship between ITS Amplicon and 16S

The relationship between fungal (ITS) and bacterial (16S) samples was examined. We decided to expose only the mean between both time points, which was significant for the Chao1 index (R^2^ = 0.18, *p* < 0.01) (Figure 4A). The trend was also similar for the Shannon index (R^2^ = 0.04, *p* = 0.181) (Figure 4B).

Other analyses performed to detect associations between fungal classes and the most abundant bacterial genera did not reveal joint positive or negative variations after adjusting for FDR (Appendix A). A decrease in the fungal class, Agaricomycetes, was observed with an increase in the bacterial genus *Roseburia* post-MD (Spearman’s correlation, *p* < 0.01; FDR = 0.067).

### 3.4. Gastrointestinal Disorders

Almost all the GI disorders presented by the participants decreased post-MD (Table 2). The MD had the greatest impact on diarrhea, and its intensity decreased from moderate to mild. Flatulence, bloating, and abdominal pain present at higher intensities in the pre-MD participants were significantly reduced post-MD (FDR < 0.05), except for flatulence, which showed results close to the limit of significance adjusted for multiple comparisons (*p* < 0.05; FDR = 0.088). Post-MD participants also reported improvements in nausea, burning sensations, acid regurgitation, and epigastric pain. MD had no effect on the remaining symptoms after six weeks of intervention.

To improve our understanding of GI disorders and biomarkers, correlations with the most abundant fungal classes and bacterial genera pre-MD were examined. In the first case, it was only observed that nausea increased depending on the Agaricomycetes fungal class (Appendix A), while positive associations were found between the bacterial genera *Alistipes* and *Roseburia* with constipation and diarrhea, respectively, and the urge to defecate and diarrhea were negatively associated with *Alistipes* (Figure 5A). Correlations among GI disorders, biomarkers, age, sex, and BMI were also studied (Figure 5B). Numerous symptoms were predictably positively correlated given their intuitive covariation. There was a relationship between constipation and intestinal permeability measured through zonulin, the results of which were oriented in the same direction as those observed for constipation and zonulin in the bacterial genus *Alistipes*.

### 3.5. Biomarkers

Most biomarkers (iron, ferritin, transferrin and zonulin) decreased slightly post-MD, without reaching significance (Table 2). Regarding the associations between biomarkers and bacterial genera, *Alistipes* showed a strong positive association with zonulin, and *Eubacterium* was the only genus that was positively associated with transferrin in the opposite direction (Figure 5A). Post-MD visualization was omitted because only zonulin was positively associated with *Bacteroides* following a positive pre-MD trend (FDR < 0.05). Furthermore, these biomarkers were included in the correlations described in the previous section (Figure 5B).

## 4. Discussion

This study aimed to investigate the effects of a six-week MD after intervention on the composition and diversity of bacterial and fungal gut microbiota as well as on digestive system symptoms and biomarkers in people with GI disorders. These findings suggest that MD may change the composition and diversity of intestinal fungi at phyla and genera level, while the bacterial community remains unchanged. Biodiversity decreased radically in fungi, particularly in Chytridiomycota, Mucoromycota, Sordariomycetes, Leotiomycetes, Rhizophydiomycetes, Spizellomycetes, Umbelopsidomycetes, and Orbiliomycetes. In contrast, the abundance of Saccharomycetes increased post-MD. Specifically, *Saccharomyces* seems to be positively associated with dairy consumption and inversely associated with *Candida* [39]. Shuai et al. observed that abundance of *Saccharomyces* was positively associated with decanoic acid, which has an antibacterial and an anti-inflammatory role, and with high-density lipoprotein cholesterol (c-HDL), whereas it was inversely associated with fasting glucose. In fact, the use of probiotic based on some species of Saccharomyceres class (such as *Saccharomyces cerevisiae* var. *boulardii)* is a good treatment against different acute and chronic gastrointestinal diseases [40]. However, it has been observed that the abundance of *Saccharomyces* could be also higher in patients with IBD [41]. These contradictory results could be explained due to initial studies on gut fungi were limited to species that could be isolated and cultured [42], although further studies are needed to confirm it. The association between Eurotiomycetes and Leotiomycetes classes and sex was also revealing, being higher in men than in women (with a smaller effect size), along with a reduction in the Basidiomycota/Ascomycota ratio. Some authors have shown changes in the fungal intestinal microbiota, with significant differences in variation in fungal species between the sexes, with richness and abundance being greater in women than in men. Analysis at the sex level showed that *Penicillium, Aspergillus*, and *Candida* were the most abundant genera, being *Aspergillus* and Tremellomycetes significantly more abundant in men than in women [43]. This can be attributed to the role of sex hormones in modulating microbiota composition [44,45].

Fungal data contrast with intestinal bacterial community, which was not altered post-MD. In this sense, the reported results are inconclusive given the sample variability and intervention time. Some studies have reported an increase in bacterial diversity in elderly people assessed one year following MD in different European countries [46], and a tendency to increase was also found in adults with high adherence to MD [47]. In the meta-analysis carried out by Illescas et al. (2021) [48], MD appears to promote an increase in *Akkermansia* (a marker of a healthy gut) and a reduction in *Fusobacterium* (a pathogenic bacteria). These benefits could be the result of the main components of the MD (dietary fibers, polyphenols, polyunsaturated fatty acids ω-3, and other micronutrients) [6]. Meanwhile, other studies observed no appreciable changes in the bacterial diversity of patients with obesity and/or metabolic syndrome after one year of intervention [49] or in healthy patients subjected to a fast-food crossover design for four days [50]. Both studies [49,50] reported differential abundances according to the MD. In fact, a recent review also shows that there is no clear evidence of MD effect on gut microbiota [51]. However, a large portion of the bacterial composition changes reported in the scientific literature can be explained by the different methods used and the disparity in the results they offer [51,52]. Regarding the Firmicutes/Bacteroidetes ratio, a reduction was observed in our study that did not reach significance and, in this sense, variable results have been obtained in previous investigations. While some report that the ratio is lower in people with greater adherence to MD [47], others do not appreciate the changes [53].

According to the analyses of the two amplicons, there were no joint variations in the fungal classes and bacterial genera. There was a slight decrease (without reaching significance) in the fungal class Agaricomycetes with an increase in the bacterial genus *Roseburia*. These changes could be explained due to competition between fungi and bacteria. It has been observed that some bacteria can inhibit the growth of fungi through a modification of the environment and available nutrients [54].

Regarding the assessment of MD adherence, analyzes using MEDAS questionnaire scores were unsuccessful and provided conflicting data. The positive trend in richness observed with the Chao1 index before MD with high MD adherence scores does not appear to be consistent with post-MD dysbiosis. To date, few studies [47,55,56,57] have investigated human microbiota using the MEDAS questionnaire with bacterial samples and none have investigated fungi. Therefore, further research is needed on this topic.

A general improvement of post-MD GI disorders was observed in our study, which highlights the benefits previously reported by three diets (Low-fermentable oligosaccharides, disaccharides, monosaccharides, and polyols (FODMAP); gluten-free; and balanced) on irritable bowel syndrome (IBS) symptoms [58]. The biomarker values (iron, ferritin, transferrin and zonulin) also decreased in most post-MD measurements; however, the decrease was not sufficient to confirm the changes between time points. In particular, the reduction in zonulin was greater than that observed in the other measurements. In line with this finding, a recent study [59] reported lower zonulin concentrations in >100 women with intestinal barrier deterioration who underwent the MD for three months. These differences could be due to the selected sample or power of the study, among other factors, which could have reduced the expected change in our study. However, other investigators [60] reported that patients with impaired intestinal permeability remained the same after 16 weeks of the MD, as measured using the chromium-51 ethylenediamine tetraacetate excretion test (51Cr-EDTA).

The impact of iron on microbiota composition, health, and pathogenesis of intestinal inflammatory diseases has received considerable attention [61,62]. Based on these results, *Eubacterium* was the only genus negatively associated with transferrin. The latter pattern is contrary to that observed in an in vitro colonic fermentation study using immobilized human fecal microbiota to demonstrate the impact of iron deficiency. The data showed that several taxa, including *Roseburia, Eubacterium, Clostridium*, and *Bacteroides* decreased, whereas members of the *Lactobacillus* and Enterobacteriaceae families increased [63].

The role of bacteria in digestive system symptoms and biomarkers is also relevant. Symptoms related to stool consistency and zonulin levels pre-MD were associated with the bacterial genera *Alistipes* and *Roseburia*, as well as *Alistipes* with zonulin. These results suggest that *Alistipes* and *Roseburia* play important roles in intestinal transit. The genus *Roseburia* is a commensal bacterium that produces SCFAs such as butyrate, which affects colon motility [64]. This trend has been observed in women with rapid transit [65] as well as its lower abundance in the intestines of adults and children [66,67,68]. In addition, a significant decrease in butyrate concentration has been observed in patients who suffer from inflammatory bowel disease, irritable bowel syndrome, and colon cancer [65]. This fact could be a consequence of a decrease in *Roseburia spp*, among others [65]. However, the relationship between *Alistipes* and intestinal transit remains unclear. While no differences were the most common finding, a higher abundance of *Alistipes* has been reported in constipated people than in healthy controls [67,69] and vice versa [70]. Other studies have been associated the genus *Alistipes* with protective effects against gastrointestinal diseases, such as colitis, which is associated with diarrhea episodes [71]. To the best of our knowledge, no studies have analyzed intestinal permeability together with the intestinal microbiota to compare our findings with those of the genus *Alistipes*. Therefore, incorporating zonulin measurements into future research studies is encouraged.

Interestingly, constipation was positively associated with zonulin levels. Patients with IBS with diarrhea (IBS-D) or constipation (IBS-C) have already been reported to have higher serum zonulin levels than healthy controls [72,73], although other studies have found no differences in patients with IBS-D [74,75]. Singh et al. [72] observed that zonulin levels were positively correlated with stool frequency in patients with IBS-D, unlike the results of this study, whereas Ohlsson et al. [76] found no correlation between diarrhea or constipation and serum zonulin levels.

Among all GI disorders, burning sensations was negatively associated with zonulin levels in this study. None of the other symptoms correlated with intestinal permeability, as in previous studies measuring zonulin levels [76] or using the three-sugar test [77,78]. A positive correlation has been found between the severity of bloating and zonulin levels in patients with IBS, specifically those with IBS-D [72].

The absence of associations between symptoms, bacteria, and zonulin levels post-MD was found, and there was no significant reduction in the composition of *Alistipes* genera and in post-MD zonulin levels. However, patients still improved most of their GI symptoms, which could indicate some specific mediation or association between the bacteria, intestinal permeability, and fungi that could not be explained.

The research design was limited to two time points (pre- and post-MD) and did not include a control group. Therefore, the extrapolation of the results to the study population and the causal relationship between the MD and changes observed in the gut microbiota are questionable. The initial research strategy was to classify participants with low adherence to the MD as the experimental group and participants with high adherence to the MD as the control group. This could not be done because the scores obtained through the MEDAS were unable to explain not only the dysbiosis occurring in the fungal community but also any phenomenon analyzed. The MEDAS, whose sum of scores per participant was obtained through dichotomous responses, did not satisfied the desire to accurately measure adherence to the MD. In fact, the relationship between MEDAS scores and the observed fungal richness seemed positive pre-MD contradicted the change experienced in intestinal fungi post-MD. The quality of the questionnaire in the context of analyzing adherence to the MD with the gut microbial community and, although other studies reported satisfactory results with this tool [55,56,57], the use of methods that quantify the frequency and grams of each food ingested, such as the Food Frequency Questionnaire [79].

Another decisive aspect that reduced the aspirations to report on the changes produced at different fungal taxonomic levels and the search for specific relationships of these with symptoms and biomarkers concerns the sequencing of the ITS amplicon and the reduced taxonomic assignment or number of ZOTUs at different levels, especially those subsequent to the class. This is because there was only one hegemonic class, Saccharomycetes, followed by Eurotiomycetes. There are several reasons for this, one of which is that ITS vary in length among fungi, which is a potential source of bias in the identification of species when NGS is used. Another reason is that STI variation is often insufficient to discriminate between species. Lastly, there is a lack of reference databases that present quality sequences in sufficient numbers to comprehensively identify the organisms represented [80]. This problem could raise suspicions about a possible technical cause in the identification of taxa with significant post-MD changes in fungi. However, far from being the product of sequencing and/or bioinformatic analysis, taxonomic identification was relatively similar pre- and post-MD, only between the times there was a dysbiosis which cause could be due to the MD. Nevertheless, sequencing the ITS-2 region could be a viable alternative to obtain sufficient taxonomic information in future studies [81].

In conclusion, the findings of the present study suggest that the fungal gut microbiota could be altered by MD, affecting the improvement experienced in GI disorders. This lays the groundwork for addressing the effects of dietary interventions on gut fungi and offers practical suggestions aimed at remedying the limitations of the present study to increase our understanding of fungal dysbiosis and its effects on human health.

## Figures and Tables

**Figure 1 microorganisms-12-01969-f001:**
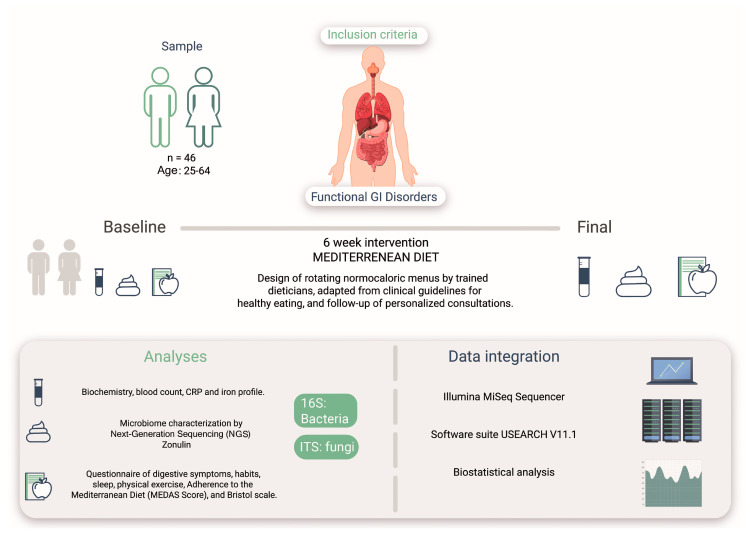
Study design based on an intervention with the MD on 46 patients presenting GI disorders. The design included analysis of bacterial and fungal microbiota, biomarkers, questionnaires on digestive symptoms, and adherence to pre- and post-MD (MEDAS).

**Figure 2 microorganisms-12-01969-f002:**
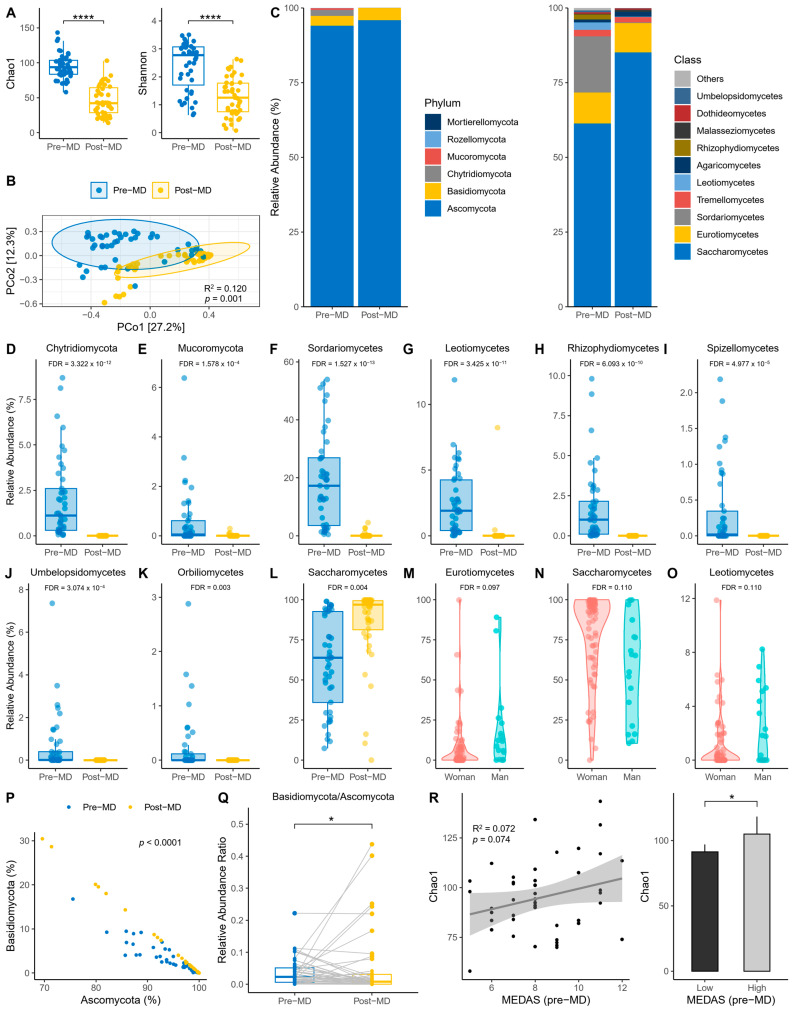
Characteristics of intestine fungi in pre- and post-MD participants. (**A**) Chao1 (left) and Shannon (right) alpha diversity indices. (**B**) Principal coordinate analysis (PCoA) based on Bray–Curtis distance. (**C**) Relative abundance of fungal phyla (left) and top 10 fungal classes (right). (**D**,**E**) Relative abundance of taxa with significant pre- and post-MD changes for phylum level. (**F**–**L**) Relative abundance of taxa with significant pre- and post-MD changes for class level. (**M**–**O**) Relative abundance of taxa with significant changes according to sex. (**P**) Relationship between Ascomycota and Basidiomycota. (**Q**) Basidiomycota/Ascomycota relative abundance ratio. (**R**) Chao1 richness based on pre-MD MEDAS score (left) and low (≤9) and high (>9) adherence categories to MD (right). The visualizations without the transformation and normalization of the data made from MaAsLin 2 are presented to improve the visibility of the extinct taxa post-MD, considering significant the values FDR < 0.05 for time and FDR < 0.15 for sex. MD, Mediterranean Diet; FDR, False Discovery Rate; NS, Not Significant; error bars represent 95% CI; significant value: * *p* < 0.05; **** *p* < 0.0001.

**Figure 3 microorganisms-12-01969-f003:**
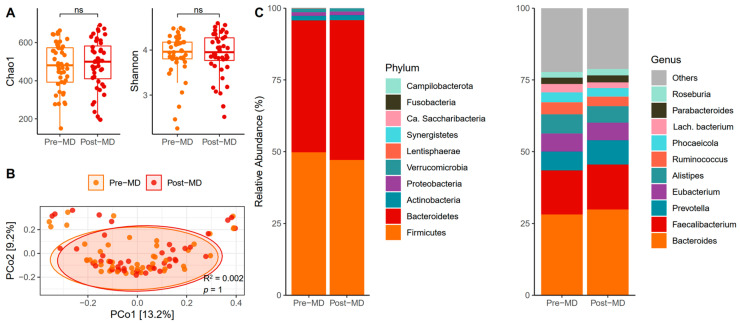
Characteristics of bacteria in the gut of pre- and post-MD participants. (**A**) Chao1 (left) and Shannon (right) diversity alpha indices. (**B**) Principal coordinate analysis (PCoA) based on Bray–Curtis distance. (**C**) Relative abundance of fungal phyla (left) and top 10 fungal classes (right). MD, Mediterranean Diet. NS, Not Significant.

**Figure 4 microorganisms-12-01969-f004:**
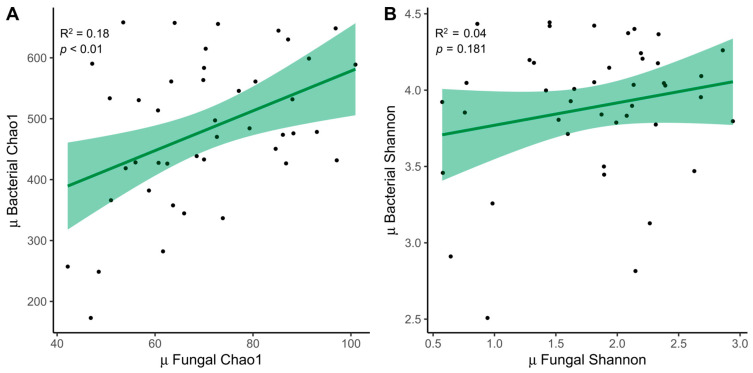
Relationship between fungi and bacteria. (**A**) Linear regressions of fungal and bacterial diversity with the mean between times for the Chao1 index and (**B**) the Shannon index. MD, Mediterranean Diet.

**Figure 5 microorganisms-12-01969-f005:**
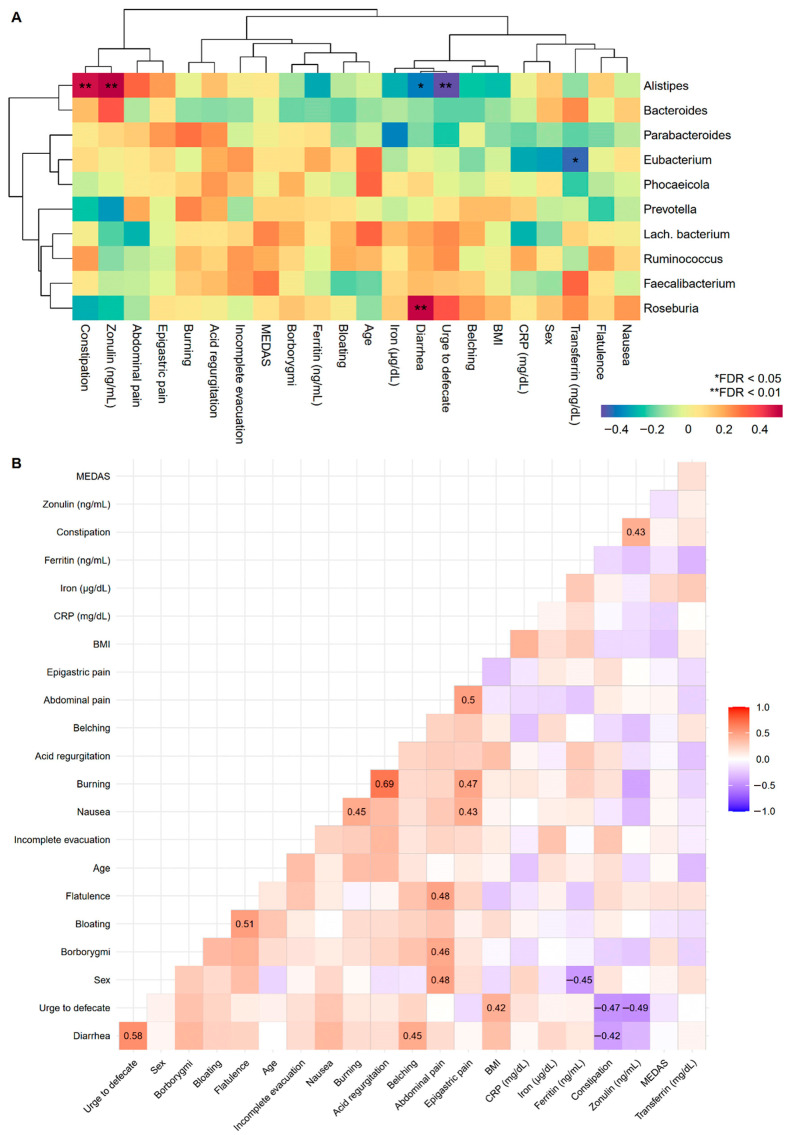
Associations between GI disorders, biomarkers, sample characteristics, and the top 10 pre-MD bacterial genera. (**A**) Hierarchical grouping of correlations with Spearman’s test between study variables and pre-MD bacterial genera. (**B**) Correlogram with Spearman’s proof between study variables. In the correlogram only the significant coefficients are presented (FDR < 0.05). The colour spectrum denotes negative (blue) to positive (red) associations. GI disorders were evaluated with ordinal response variables (0 = Absence; 1 = Mild; 2 = Moderate; 3 = Severe). Sex was coded as numerical variable (0 = Man and 1 = Woman) denoting positive associations with women and negative associations with men. CRP, C-Reactive Protein; FDR, False Discovery Rate.

**Table 1 microorganisms-12-01969-t001:** Patients’s characteristics pre-MD.

	M ± SD
Age (years)	41.39 ± 8.94
BMI (kg/m^2^)	25.41 ± 5.42
MEDAS (1–14)	8.33 ± 1.90
Sex	
Female	36 (78.3)
Male	10 (21.7)
Sleep habits	
Hours of sleep (weekday)	6.96 ± 1.17
Hours of sleep (weekend)	7.52 ± 1.23
Sleep quality (0–10)	5.96 ± 1.63
Physical activity	
Light (h/week)	0.76 ± 2.17
Moderate (h/week)	2.33 ± 2.58
Intense (h/week)	0.13 ± 0.89
MEDAS (1–14)	
Low (≤9)	33 (71.7)
High (>9)	13 (28.3)
Gastrointestinal symptoms	n (%)
Abdominal pain	42 (91.3)
Bloating	42 (91.3)
Flatulence	43 (93.5)
Diarrhea	34 (73.9)
Borborygmi	42 (91.3)
Constipation	22 (47.8)
Urge to defecate	35 (76.1)
Incomplete evacuation	35 (76.1)
Nausea	26 (56.5)
Burning sensations	29 (63)
Belching	26 (56.5)
Acid regurgitation	27 (58.7)
Epigastric pain	30 (65.2)

M ± SD, Mean ± Standard Desviation; n, frecuency. BMI: Body Mass Index; MEDAS: The Mediterranean Diet Adherence Screener.

**Table 2 microorganisms-12-01969-t002:** GI disorders, blood biomarkers, and intestinal permeability pre- and post-MD.

	Pre-MD (n = 46)	Post-MD (n = 46)	*p*	FDR
GI disorders				
Abdominal pain	5.8 ± 3.02	4.42 ± 2.98	<0.05	<0.05
Bloating	7.03 ± 3.08	5.51 ± 3.46	<0.01	<0.05
Flatulence	7.17 ± 2.98	6.30 ± 3.08	<0.05	0.088
Diarrhea	5.29 ± 3.95	2.90 ± 3.41	<0.0001	<0.0001
Stomach sounds	5.22 ± 3.03	4.49 ± 3.39	0.115	0.188
Constipation	2.32 ± 2.97	2.32 ± 3.43	1	1
Urge to defecate	4.85 ± 3.63	3.77 ± 3.95	<0.05	0.069
Incomplete evacuation	4.64 ± 3.40	3.77 ± 3.49	0.070	0.126
Nausea	2.83 ± 2.89	1.81 ± 2.78	<0.01	<0.05
Burning sensations	3.62 ± 3.36	2.25 ± 2.81	<0.01	<0.05
Belching	3.33 ± 3.58	2.90 ± 3.34	0.309	0.428
Acid regurgitation	2.68 ± 2.86	1.74 ± 2.79	<0.05	<0.05
Epigastric pain	4.20 ± 3.54	2.61 ± 3.29	<0.01	<0.05
Biomarkers				
Iron (μg/dL)	95.67 ± 36.48	91.39 ± 38.75	0.511	0.613
Ferritin (ng/mL)	85.55 ± 78.07	83.46 ± 82.43	0.769	0.865
Transferrin (mg/dL)	249.01 ± 47.61	245.78 ± 38.41	0.465	0.598
CRP (mg/dL)	0.24 ± 0.30	0.25 ± 0.31	0.841	0.890
Zonulin (ng/mL)	124.15 ± 124.8	101.62 ± 66.52	0.220	0.330

M ± SD, Mean ± Standard Desviation; FDR, False Discovery Rate. CRP, C-Reactive Protein. GI disorders were evaluated using ordinal response variables (0 = absent; 1 = mild; 2 = moderate; 3 = severe) whose scores were standardized to 10.

## Data Availability

The data presented in this study are available from the corresponding author on reasonable request due to privacy and identification of the patients.

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
