# Peer review of "Mediterranean Diet Effect on the Intestinal Microbiota, Symptoms, and Markers in Patients with Functional Gastrointestinal Disorders"

_microorganisms, 2024, doi:10.3390/microorganisms12101969_

Round 1

Reviewer 1 Report

Comments and Suggestions for Authors

This peer-reviewed manuscript focuses on gut mycobiome in patients on a specific diet. Studies on fungal diversity have recently been actively conducted and are at the stage of data accumulation and analysis. The main problem in this area of research is methodological; the approach that is effectively used to study bacteria — amplification of variable regions of the ribosomal operon — turned out to be less effective when applied to fungi. Given this problem, many authors repeatedly write that the methodological approach and its correct implementation are important. Obviously, the main requirement for the submitted manuscript is that the methodological part of the proposed work be written correctly and in sufficient detail. The comments are systematized as follows. Considering that the manuscript has been submitted to the Journal named "Microorganisms", the additional information I requested is necessary.

1. The introduction should be carefully reviewed and updated. The authors present references from 8 and/or 10 years ago as arguments proving the low level of fungal microbiota investigations. The Introduction does not contain the main relevant publications on the given topic for 2023-2024 (the only one publication of 2023 is presented). This situation is absolutely not indicative because in the last 3 years (2022-2024) the interest and publication activity in this area of research has significantly increased.

2. The authors use an old taxonomy of bacteria throughout the text of the manuscript; therefore, it is necessary to carefully proofread the manuscript and make corrections in accordance with the current taxonomy. See https://www.bacterio.net/ and Oren A, Garrity GM. Valid publication of the names of forty-two phyla of prokaryotes. Int J Syst Evol Microbiol 2021; 71:5056.

3. Section 2.6.3. DNA library construction: The authors presented the sample preparation for shotgun sequencing, but in section 2.6.4. Sequencing they mentioned amplicon sequencing for both bacteria and fungi. It is necessary to describe the method used to obtain sequencing data more clearly. Primers used in the study, amplification conditions, and kit used for sequencing. In addition, all primary data should be deposited in an open database, and patients can be anonymized. I cannot verify the accuracy of the results provided by the authors.

The primary results of this study were derived from sequencing data. The methods for obtaining these data should be described more clearly and with appropriate links to the original sources and electronic resources.

4. Section 2.7. Bioinformatic analysis should be supplemented with information on how the primary processing of sequencing data was performed; for example, what was done with single and doubletons and were they excluded from the analysis or considered rare sequences? This affected, at least, the results of the alpha diversity assessment. Based on the information provided by the authors in the methodological section, it was impossible to assess the accuracy of their analysis.

5. L. 209: The amplicon sequence data were not rarefied because the sequencing depth was consistent across all samples. Visualization of these data is required. It is not clear whether the authors received sufficient data from each patient.

6. L. 216-217: What does this mean? 80% is a very small percentage if the remaining 20% are unidentified sequences. However, several phylotypes have recently been identified as belonging to a certain family or order, and they are named UCG-XX. Their homologs were found in different habitats, suggesting their ubiquity and temporarily unidentified position. Conversely, some sequences have a low percentage of homology with the type species of a certain genus, and taxonomy algorithms identify them as undefined. I recommend that the authors consider widely used algorithms for taxonomic assignment to bacteria.

This manuscript requires serious revision and cannot be recommended for publication in its current form.

Author Response

EVALUATION:
This peer-reviewed manuscript focuses on gut mycobiome in patients on a specific diet. Studies on fungal diversity have recently been actively conducted and are at the stage of data accumulation and analysis. The main problem in this area of research is methodological; the approach that is effectively used to study bacteria — amplification of variable regions of the ribosomal operon — turned out to be less effective when applied to fungi. Given this problem, many authors repeatedly write that the methodological approach and its correct implementation are important. Obviously, the main requirement for the submitted manuscript is that the methodological part of the proposed work be written correctly and in sufficient detail. The comments are systematized as follows. Considering that the manuscript has been submitted to the Journal named "Microorganisms", the additional information I requested is necessary.

Comment 1: The introduction should be carefully reviewed and updated. The authors present references from 8 and/or 10 years ago as arguments proving the low level of fungal microbiota investigations. The Introduction does not contain the main relevant publications on the given topic for 2023-2024 (the only one publication of 2023 is presented). This situation is absolutely not indicative because in the last 3 years (2022-2024) the interest and publication activity in this area of research has significantly increased.

Response 1: Thank you for your comment. Several new references have been included in the introduction section (publications between 2021-2024).

Comment 2: The authors use an old taxonomy of bacteria throughout the text of the manuscript; therefore, it is necessary to carefully proofread the manuscript and make corrections in accordance with the current taxonomy. See https://www.bacterio.net/ and Oren A, Garrity GM. Valid publication of the names of forty-two phyla of prokaryotes. Int J Syst Evol Microbiol 2021; 71:5056.

Response 2: Thank you very much for your comment. It is not possible to change the taxonomy due to bioinformatics analysis was carried out by Silva (https://www.arb-silva.de/) in 2023. Silva is an external laboratory and they set up the libraries.

Comment 3: Section 2.6.3. DNA library construction: The authors presented the sample preparation for shotgun sequencing, but in section 2.6.4. Sequencing they mentioned amplicon sequencing for both bacteria and fungi. It is necessary to describe the method used to obtain sequencing data more clearly. Primers used in the study, amplification conditions, and kit used for sequencing. In addition, all primary data should be deposited in an open database, and patients can be anonymized. I cannot verify the accuracy of the results provided by the authors.

The primary results of this study were derived from sequencing data. The methods for obtaining these data should be described more clearly and with appropriate links to the original sources and electronic resources.

Response 3: Thank you for your assessment. Silva (https://www.arb-silva.de/), an external laboratory, carried out bioinformatics analysis. We cannot publish their specific procedures because this information is part of their confidentiality. However, we will share again the raw dataset. 

Comment 4: Section 2.7. Bioinformatic analysis should be supplemented with information on how the primary processing of sequencing data was performed; for example, what was done with single and doubletons and were they excluded from the analysis or considered rare sequences? This affected, at least, the results of the alpha diversity assessment. Based on the information provided by the authors in the methodological section, it was impossible to assess the accuracy of their analysis.

Response 4: Thank you for your comment. As usual, this type of anomalous data is removed from the analysis and statistics.

Comment 5: L. 209: The amplicon sequence data were not rarefied because the sequencing depth was consistent across all samples. Visualization of these data is required. It is not clear whether the authors received sufficient data from each patient.

Response 5: Of course, the bioinformaticians presented the information to us in the following visualizations:

Rarefaction curves for each study sample showing the number of reads versus the number of detected ZOTUs (left). Representation of the frequencies of ZOTUs by comparing the number of observed ZOTUs in each sample against their expected (theoretical) value using the rarefaction technique applied to a certain number of reads (right).

They provided us with this data for a broader longitudinal study that included a third time point, and whose results are unrelated to the effects of MD.

Comment 6: L. 216-217: What does this mean? 80% is a very small percentage if the remaining 20% are unidentified sequences. However, several phylotypes have recently been identified as belonging to a certain family or order, and they are named UCG-XX. Their homologs were found in different habitats, suggesting their ubiquity and temporarily unidentified position. Conversely, some sequences have a low percentage of homology with the type species of a certain genus, and taxonomy algorithms identify them as undefined. I recommend that the authors consider widely used algorithms for taxonomic assignment to bacteria.

Response 6: We believe that achieving 80% taxonomic assignment is satisfactory and aligns with current standards. We are also aware that there are numerous microbiome analysis suites that contain different algorithms, but the bioinformatics analysts recommended using USEARCH, and it is not feasible for us to rerun the taxonomic assignment with other algorithms.

Reviewer 2 Report

Comments and Suggestions for Authors

This manuscript presents a study on the impact of the Mediterranean diet on the intestinal microbiota and its association with gastrointestinal symptoms and biomarkers in patients with functional gastrointestinal disorders. 

The main flaws are the study design:

(1) a control group is lacked in the study design, eventhough it is somehow difficult to set a control group in such study, maybe a patient group without MD treatment?

(2) is six-week intervention long enough to alter the bacterial composition, could it be the reason of the unchangged bacterial community?

(3) it is unclear if the patient was informed the aim of the study they were involved in, this is important since IT may chang the results; in a more scientific design, a group similar to “placebo group” should be set.

minor issues:

(1) Table 1: the sex distribution indicates a significant predominance of females over males. Could you please elucidate whether this disparity was intentional, and if so, what was the rationale behind this specific selection criterion?

(2) Figure 1: “next-generation sequencing” was designated as “NSG”, I believe it should be “NGS”.

(3) the most interesting finding is that the MD treatment altered the fungal composition instead of bacterial composition. Potienal reasons of such results, relationships of gut fungi to human GI health, especially the siganificantly-increased Saccharomycetes, should be deeply discussed.

Comments on the Quality of English Language

none.

Author Response

EVALUATION:
This manuscript presents a study on the impact of the Mediterranean diet on the intestinal microbiota and its association with gastrointestinal symptoms and biomarkers in patients with functional gastrointestinal disorders.

The main flaws are the study design:

Comment 1: a control group is lacked in the study design, eventhough it is somehow difficult to set a control group in such study, maybe a patient group without MD treatment?

Response 1: Thank you very much for the information. We are aware of this limitation and will take it into account for future studies.

Comment 2: is six-week intervention long enough to alter the bacterial composition, could it be the reason of the unchangged bacterial community?

Response 2: Thank you for your question. In just 6 weeks of dieting, we know that the bacterial composition of the gut microbiota can be significantly altered. Although we didn’t observe changes in this regard, we were able to find improvements in the patients' symptoms.

Comment 3: it is unclear if the patient was informed the aim of the study they were involved in, this is important since IT may chang the results; in a more scientific design, a group similar to “placebo group” should be set.

Response 3: All patients were informed about the objectives of the study. Related “placebo group”, we will take it into account for future studies.

Minor issues:

Comment 1: Table 1: the sex distribution indicates a significant predominance of females over males. Could you please elucidate whether this disparity was intentional, and if so, what was the rationale behind this specific selection criterion?

Response 1: Thank you for your comment. This distribution was randomized. Patients who came to consult on those dates and satisfy selection criteria were mostly women. We would have liked there to be a better balance between both sexes, but it wasn't possible.

Comment 2: Figure 1: “next-generation sequencing” was designated as “NSG”, I believe it should be “NGS”.

Response 2: Thank you for bringing this to our attention. Figure 1 has been corrected.

Comment 3: the most interesting finding is that the MD treatment altered the fungal composition instead of bacterial composition. Potienal reasons of such results, relationships of gut fungi to human GI health, especially the siganificantly-increased Saccharomycetes, should be deeply discussed.

Response 3: Thank you for your comment. Several new references have been included in discussion section related to Saccharomycetes.

Round 2

Reviewer 1 Report

Comments and Suggestions for Authors

I thank the authors of the manuscript for their comments; however, I do not agree with some of them and insist on finalizing the article.

Comment 2: The authors use an old taxonomy of bacteria throughout the text of the manuscript; therefore, it is necessary to carefully proofread the manuscript and make corrections in accordance with the current taxonomy. See https://www.bacterio.net/ and Oren A, Garrity GM. Valid publication of the names of forty-two phyla of prokaryotes. Int J Syst Evol Microbiol 2021; 71:5056.

Response 2: Thank you very much for your comment. It is not possible to change the taxonomy due to bioinformatics analysis was carried out by Silva (https://www.arb-silva.de/) in 2023. Silva is an external laboratory, and they set up the libraries.

Nothing is impossible, and the authors need to make changes to the names of the phyla in the main text of the article according to the recommendations https://www.bacterio.net/ and Oren A, Garrity GM. Valid publication of the names of forty-two phyla of prokaryotes. Int J Syst Evol Microbiol 2021; 71:5056. In addition, if the authors do not use alternative data visualization algorithms, they should provide appropriate explanations in the figure captions. I would like to remind you that the article has been submitted to Microorganisms. It is assumed that articles should contain up-to-date information, primarily on the taxonomy of microorganisms.

Comment 3: Section 2.6.3. DNA library construction: The authors presented the sample preparation for shotgun sequencing, but in section 2.6.4. Sequencing they mentioned amplicon sequencing for both bacteria and fungi. It is necessary to describe the method used to obtain sequencing data more clearly. Primers used in the study, amplification conditions, and kit used for sequencing. In addition, all primary data should be deposited in an open database, and patients can be anonymized. I cannot verify the accuracy of the results provided by the authors.

The primary The results of this study were derived from sequencing data. The methods for obtaining these data should be described more clearly and with appropriate links to the original sources and electronic resources.

Response 3: Thank you for your assessment. Silva (https://www. arb-silva.de/), an external laboratory, carried out bioinformatics analysis. We cannot publish their specific procedures because this information is part of their confidentiality. However, we will share again the raw dataset.

Strongly disagree. It is necessary, that Section 2.6 adequately describes the methods used by the authors. The current presentation of the methods is incorrect. For example, the description in L. 168-169 assumes that the authors obtained metagenomic libraries, but these data were not included in the manuscript. Based on L. 178, the authors prepared amplicon libraries. Methods should be presented correctly.

Comment 4: Section 2.7. Bioinformatic analysis should be supplemented with information on how the primary processing of sequencing data was performed; for example, what was done with single and doubletons and were they excluded from the analysis or considered rare sequences? This affected, at least, the results of the alpha diversity assessment. Based on the information provided by the authors in the methodological section, it was impossible to assess the accuracy of their analysis.

Response 4: Thank you for your comment. As usual, this type of anomalous data is removed from the analysis and statistics.

This information is critical and should be included in the methods.

Comment 5: L. 209: The amplicon sequence data were not rarefied because the sequencing depth was consistent across all samples. Visualization of this data is required. It is not clear whether the authors received sufficient data from each patient.

Response 5: Of course, the bioinformaticians present the information to us in the following visualizations:

Rarefaction curves for each study sample showing the number of reads versus the number of detected ZOTUs (left). Representation of the frequencies of ZOTUs by comparing the number of observed ZOTUs in each sample against their expected (theoretical) value using the rarefaction technique applied to a certain number of reads (right).

They provided us with this data for a broader longitudinal study that included a third time point, and whose results are unrelated to the effects of MD.

The information is informative. Include this information in the manuscript, perhaps as an additional material.

Additional note. I looked through the references cited by the authors and found duplications; in connection with this, I recommend that the authors once again carefully check the correspondence of the references. Please delete these duplications:

L. 562-563: 5. Guasch-Ferré, M.; Willett, W.C. The Mediterranean diet and health: a comprehensive overview. J. Intern. Med. 2021, 290: 549–566. doi: 10.1111/joim.13333.

L. 624-625: 33. Guasch-Ferré, M.; Willett, W.C. The Mediterranean diet and health: a comprehensive overview. J. Intern. Med. 2021, 290: 549–566. doi: 10.1111/joim.13333.

L. 699-700: 65. Tamanai-Shacoori, Z.; Smida, I.; Bousarghin, L.; Loreal, O.; Meuric, V.; Fong, S.B. et al. Roseburia spp.: a marker of health? Future Microbiol. 2017, 12: 157–170. doi: 10.2217/fmb-2016-0130.

L. 712-713: 70. Tamanai-Shacoori, Z.; Smida, I.; Bousarghin, L.; Loreal, O.; Meuric, V.; Fong, S.B. et al. Roseburia spp.: a marker of health? Future Microbiol. 2017, 12:157-170. doi: 10.2217/fmb-2016-0130. 

Author Response

I thank the authors of the manuscript for their comments; however, I do not agree with some of them and insist on finalizing the article.

Comment 2: The authors use an old taxonomy of bacteria throughout the text of the manuscript; therefore, it is necessary to carefully proofread the manuscript and make corrections in accordance with the current taxonomy. See https://www.bacterio.net/ and Oren A, Garrity GM. Valid publication of the names of forty-two phyla of prokaryotes. Int J Syst Evol Microbiol 2021; 71:5056.

Response 2: Thank you very much for your comment. It is not possible to change the taxonomy due to bioinformatics analysis was carried out by Silva (https://www.arb-silva.de/) in 2023. Silva is an external laboratory and they set up the libraries.

Nothing is impossible, and the authors need to make changes to the names of the phyla in the main text of the article according to the recommendations https://www.bacterio.net/ and Oren A, Garrity GM. Valid publication of the names of forty-two phyla of prokaryotes. Int J Syst Evol Microbiol 2021; 71:5056. In addition, if the authors do not use alternative data visualization algorithms, they should provide appropriate explanations in the figure captions. I would like to remind you that the article has been submitted to Microorganisms. It is assumed that articles should contain up-to-date information, primarily on the taxonomy of microorganisms.

Response 2.1: Our apologies, it is not possible to change the taxonomy.

Comment 3: Section 2.6.3. DNA library construction: The authors presented the sample preparation for shotgun sequencing, but in section 2.6.4. Sequencing they mentioned amplicon sequencing for both bacteria and fungi. It is necessary to describe the method used to obtain sequencing data more clearly. Primers used in the study, amplification conditions, and kit used for sequencing. In addition, all primary data should be deposited in an open database, and patients can be anonymized. I cannot verify the accuracy of the results provided by the authors.

The primary results of this study were derived from sequencing data. The methods for obtaining these data should be described more clearly and with appropriate links to the original sources and electronic resources.

Response 3: Thank you for your assessment. Silva (https://www.arb-silva.de/), an external laboratory, carried out bioinformatics analysis. We cannot publish their specific procedures because this information is part of their confidentiality. However, we will share again the raw dataset. 

Strongly disagree. It is necessary, that Section 2.6 adequately describes the methods used by the authors. The current presentation of the methods is incorrect. For example, the description in L. 168-169 assumes that the authors obtained metagenomic libraries, but these data were not included in the manuscript. Based on L. 178, the authors prepared amplicon libraries. Methods should be presented correctly.

Response 3.1: Our apologies, this information will not be included in the manuscript.

Comment 4: Section 2.7. Bioinformatic analysis should be supplemented with information on how the primary processing of sequencing data was performed; for example, what was done with single and doubletons and were they excluded from the analysis or considered rare sequences? This affected, at least, the results of the alpha diversity assessment. Based on the information provided by the authors in the methodological section, it was impossible to assess the accuracy of their analysis.

Response 4: Thank you for your comment. As usual, this type of anomalous data is removed from the analysis and statistics.

This information is critical and should be included in the methods.

Response 4.1: Our apologies, this information will not be included in the manuscript.

Comment 5: L. 209: The amplicon sequence data were not rarefied because the sequencing depth was consistent across all samples. Visualization of these data is required. It is not clear whether the authors received sufficient data from each patient.

Response 5: Of course, the bioinformaticians presented the information to us in the following visualizations:

Rarefaction curves for each study sample showing the number of reads versus the number of detected ZOTUs (left). Representation of the frequencies of ZOTUs by comparing the number of observed ZOTUs in each sample against their expected (theoretical) value using the rarefaction technique applied to a certain number of reads (right).

They provided us with this data for a broader longitudinal study that included a third time point, and whose results are unrelated to the effects of MD.

The information is informative. Include this information in the manuscript, perhaps as an additional material.

Response 5.1: Thank you for the comment. This information has been submitted as “Supplementary File”.

Additional note. I looked through the references cited by the authors and found duplications; in connection with this, I recommend that the authors once again carefully check the correspondence of the references. Please delete these duplications:

  1. 562-563: 5. Guasch-Ferré, M.; Willett, W.C. The Mediterranean diet and health: a comprehensive overview. J. Intern. Med. 2021, 290: 549–566. doi: 10.1111/joim.13333.
  2. 624-625: 33. Guasch-Ferré, M.; Willett, W.C. The Mediterranean diet and health: a comprehensive overview. J. Intern. Med. 2021, 290: 549–566. doi: 10.1111/joim.13333.
  3. 699-700: 65. Tamanai-Shacoori, Z.; Smida, I.; Bousarghin, L.; Loreal, O.; Meuric, V.; Fong, S.B. et al. Roseburia spp.: a marker of health? Future Microbiol. 2017, 12: 157–170. doi: 10.2217/fmb-2016-0130.
  4. 712-713: 70. Tamanai-Shacoori, Z.; Smida, I.; Bousarghin, L.; Loreal, O.; Meuric, V.; Fong, S.B. et al. Roseburia spp.: a marker of health? Future Microbiol. 2017, 12:157-170. doi: 10.2217/fmb-2016-0130. 

Response additional note: Thank you very much for your comment. References have been reviewed and corrected.

Reviewer 2 Report

Comments and Suggestions for Authors

Manuscript has been revised based on the comments.

Author Response

EVALUATION:

Manuscript has been revised based on the comments.

Response: Manuscript has been revised based on your previous comments. Thank you.
